# Bidirectional Translation Between ECG and PCG

Sajjad Karimi
*Dept. of Biomed. Informatics*
*Emory University*
Atlanta, USA
sajjad.karimi@dbmi.emory.edu

Amit J. Shah
*Dept. of Epidemiology*
*Dept. of Medicine*
*Emory University*
Atlanta, USA
ajshah3@emory.edu

Gari D. Clifford
*Dept. of Biomed. Informatics*
*Emory University*
*Dept. of Biomed. Eng.*
*Georgia Tech*
Atlanta, USA
gari@gatech.edu

Reza Sameni[*]
*Dept. of Biomed. Informatics*
*Emory University*
*Dept. of Biomed. Eng.*
*Georgia Tech*
Atlanta, USA
rsameni@dbmi.emory.edu

*Abstract*—Simultaneous electrocardiography (ECG) and phonocardiogram (PCG) offer a multimodal view of cardiac function by capturing electrical and mechanical activity, respectively. However, their shared and unique information and potential for mutual reconstruction remain poorly understood across different physiological states and individuals.

This study analyzes the EPHNOGRAM dataset of simultaneous ECG-PCG recordings during rest and exercise, using linear and nonlinear models—including a non-causal neural network—to reconstruct one modality from the other. Nonlinear models, especially non-causal neural network, outperform others, with ECG reconstruction from PCG proving more feasible. In the within-subject scenario, the non-causal neural network achieved a signal-to-noise ratio (SNR) of $6.5 \pm 5.2$ dB and a cross-correlation of $0.78 \pm 0.19$ for PCG-based ECG reconstruction.

These findings provide quantitative insight into the electromechanical relationship between cardiac signals and support the development of multimodal cardiac monitoring tools.

*Index Terms*—ECG-PCG Translation, Cross-modal learning, Machine-learning, Power spectrum

## I. Introduction

Cardiac function arises from tightly coupled electrical, mechanical, hemodynamic, autonomic, and metabolic processes that generate interrelated biosignals. The electrocardiogram (ECG) reflects the heart's electrical activity, while cardiac auscultation captures mechanical events as heart sounds, recorded digitally as phonocardiogram (PCG) [1]. ECG and PCG are accessible, low-cost, and complementary modalities: ECG reveals conduction and repolarization, whereas PCG captures valve motion and blood flow. Integrating them provides a more complete view of cardiac function than either alone [2], [3].

Advances in sensors and machine learning have enabled joint ECG-PCG applications in monitoring [4], disease detection [5], and biomarker extraction [6], [7]. New hardware allows simultaneous acquisition [8], [9], and large datasets have spurred multimodal research, improving performance in arrhythmia classification [10], heart sound segmentation, and disease detection [11]. Coupled signal modeling and nonlinear features have further enhanced disease classification and risk stratification [2], [5].

This research was supported by the American Heart Association Innovative Project Award 23IPA1054351, on "developing multimodal cardiac biomarkers for cardiovascular-related health assessment."

Despite progress, fundamental questions remain: How much information is shared or exclusive between ECG and PCG? Can one modality be reliably reconstructed from the other across physiological states (e.g., rest vs. exercise) or subjects? Signal morphology and reconstruction are affected by sensor placement, variability, and noise [12], [13]. Prior studies often emphasize classification or basic reconstruction metrics (e.g., RMSE), neglecting clinical features, robustness, and directional information flow [14]. PCG lacks consistent mapping to ECG amplitude or morphology, and ECG variability due to electrode placement adds further complexity [12]. Open challenges include modeling causal/non-causal relationships, event timing (e.g., QT interval vs. S1/S2), and biomarker recoverability.

Applications of bimodal ECG-PCG modeling can include improved cardiac monitoring via signal denoising/reconstruction, analysis of causal/non-causal information flow to probe electromechanical coupling, and identifying modality-specific components [15] via reconstruction residuals that may serve as features for AI/ML models.

This study explores causal and non-causal relationships between ECG and PCG using simultaneous recordings during rest and exercise from the EPHNOGRAM dataset [16], [17]. We evaluate cross-modal signal reconstruction and cardiac event timing to quantify information transfer between electrical and mechanical signals across physiological states and individuals. Emphasis is placed on waveform reconstruction from one modality to another. Results highlight both the potential and limitations of cross-modal modeling for robust, generalizable cardiac monitoring.

Our unified framework advances multimodal cardiac integration by enabling accurate waveform reconstruction, with implications for wearable and remote monitoring technologies, especially in resource-constrained or ambulatory contexts [9]–[11].

The rest of the paper is organized as follows: Section II describes the dataset, preprocessing, modeling, and evaluation. Section III presents the results on reconstruction and spectral analysis. Section IV discusses implications and limitations, and Section V concludes the study.

## II. METHODS

### A. The EPHNOGRAM Dataset

The EPHNOGRAM dataset [16], [17], publicly available on PhysioNet, includes 68 simultaneous ECG-PCG recordings from healthy adults (age 25.4±1.9 years) during various physical activities. Data were collected using single-channel ECG and PCG stethoscopes in an indoor fitness center and include both short (30 s) and long (30 min) recordings. Activities span four scenarios: resting (supine and seated in a quiet room), treadmill walking at 3.7 km/h, treadmill stress testing using the modified Bruce protocol [18], and incremental-load bicycle stress testing. All signals are stored in WFDB format with aligned activity intensity labels, and detailed collection protocols are provided in [16], [17].

For this study, we analyzed 28 recordings selected for signal quality and protocol completion, excluding those with excessive noise or electrode disconnections as documented in the dataset metadata [17]. Our subset includes recordings from rest, walking, and both stress test conditions.

### B. Preprocessing

ECG baseline wander was corrected using a 0.2–30 Hz bandpass filter, and power-line interference (50 Hz) was removed using a second-order IIR notch filter (Q-factor = 45) with zero-phase forward-backward filtering.

PCG signals, representing zero-centered transient acoustic events, were bandpass filtered between 10–200 Hz. As PCG lacks baseline drift and power-line noise, no further correction was necessary.

For both ECG and PCG, outliers exceeding $\pm 6\sigma(t)$—where $\sigma(t)$ is the time-varying standard deviation computed over 1-minute sliding windows—were clipped. This empirically chosen threshold, based on expert review of EPHNOGRAM, balances artifact removal with the preservation of clinically important features like R-peaks. Lower thresholds distorted signal morphology, while higher ones failed to remove extreme noise. This setting is dataset-specific and not intended as a general guideline.

To address amplitude variability during stress testing, a time-adaptive normalization was applied using sliding 60-second windows to compute the local mean and standard deviation. This approach, aligned with prior literature [12], preserves local waveform characteristics while reducing inter-subject amplitude variation, enabling more robust cross-subject modeling.

Finally, all signals were downsampled from 8000 Hz to 1000 Hz to reduce computational and memory demands for subsequent machine learning and reconstruction analyses.

### C. Cross-modal Waveform Learning

ECG and PCG provide complementary views of cardiac function: ECG reflects electrical activity, while PCG captures mechanical events related to contraction and relaxation. Although the two modalities share some physiological information, each also conveys unique, modality-specific characteristics.

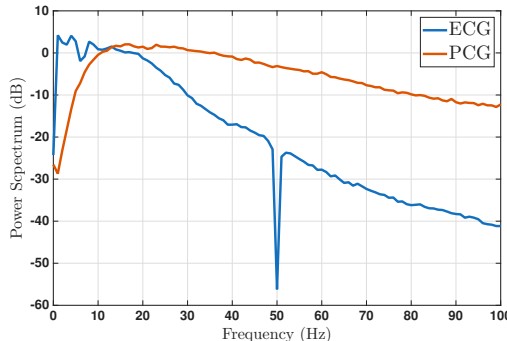

Fig. 1: Power spectrum for ECG and PCG across all recordings in EPHNOGRAM dataset.

Figure 1 illustrates this distinction through power spectral analysis of EPHNOGRAM recordings, computed via the Welch method using 1-second windows, 1 Hz resolution, and 50% overlap. The spectra reveal that low-frequency components (below 5 Hz) are primarily associated with ECG, high-frequency components (above 50 Hz) are specific to PCG, and the intermediate range (5–50 Hz) likely encodes shared electromechanical information.

*1) Machine learning frameworks:* Throughout the paper, we denote models that use one modality to predict another as $x \rightarrow y$, where $x(t)$ and $y(t)$ are time series. Accordingly, ECG→PCG refers to predicting PCG waveforms or features from ECG input using machine learning models, and PCG→ECG denotes the reverse. This prediction can be expressed as $y(t) = g(x(\tau)\,|\,t_1 \leq \tau \leq t_2)$, where $[t_1, t_2]$ is the input window of duration $\Delta t = t_2 - t_1$ used by the model $g(\cdot)$ to estimate $y$ at time $t$.

To study temporal information flow, we consider three schemes: causal, where $t_2 \leq t$ and only past input is used; anti-causal, where $t < t_1$ and only future input is used; and non-causal, where $t_1 < t < t_2$ and the input window spans both past and future. In this study, the non-causal window is symmetric around $t$, as illustrated in Fig. 2.

To explore the relationship between ECG and PCG, we employ two machine-learning models $g(\cdot)$ that span from linear to nonlinear temporal modeling. The first is a dynamic linear model using LASSO regression [19], which estimates the target modality at time $t$ from a segment of the input modality around $t$ using a sparse linear combination of samples. This model captures interpretable linear dependencies between the signals.

The second is a two-layer fully connected multilayer perceptron (MLP) with ReLU activations: the first layer has 100 neurons with a clipped ReLU, the second has 25 neurons with a leaky ReLU, and a 10% dropout layer is included to prevent overfitting. The input-output structure matches that of the linear model.

This neural network architecture was selected to balance between modeling capacity and computational efficiency. Pilot studies confirmed their robustness, eliminating the need for extensive hyperparameter tuning. Collectively, the models

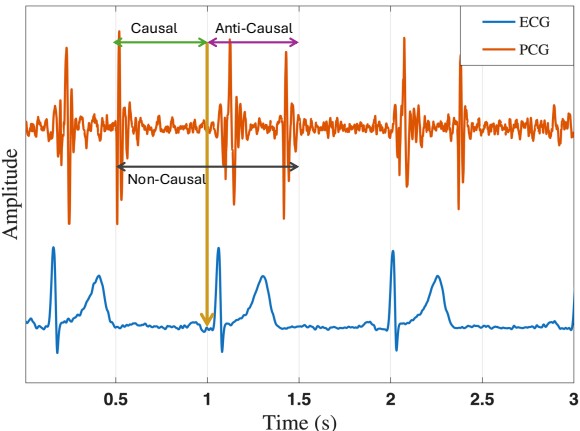

Fig. 2: Illustration of three temporal learning frameworks for transforming PCG to ECG at time $t = 1$ with a segment size of $\Delta t = 0.5$ s: (a) *causal* (green) uses only past and present inputs, (b) *anti-causal* (purple) relies on future inputs, (c) *non-causal* (black) incorporates both past and future inputs within a symmetric window.

represent a progression from simple and interpretable to complex and memory-based, allowing us to assess the modeling demands of accurate ECG-PCG reconstruction.

*2) Training and testing protocols:* We evaluate within-subject modeling to assess how well ECG and PCG information can be transferred under varying physiological conditions. Within-subject models account for intra-individual variability (e.g., fatigue).

For within-subject validation, we apply 10-fold cross-validation on each subject's 30-minute recording, using non-overlapping 3-minute segments. To avoid boundary artifacts, the first and last 1-second intervals of each recording are excluded from metric computations.

### D. Evaluation Metrics

*1) Signal-to-noise ratio:* Signal-to-noise ratio (SNR) quantifies how well a reconstructed signal matches the original waveform, treating the original ECG or PCG as the signal and the reconstruction error as noise. A higher SNR indicates better reconstruction fidelity, while an SNR of $0$ dB can result from trivial outputs (e.g., all zeros). As simultaneous ECG and PCG signals may contain physiological noise—especially during exercise—SNR should be interpreted alongside other metrics.

*2) Cross-correlation:* Cross-correlation was used to quantify the similarity between the original signal $x(t)$ and its reconstruction $\hat{x}(t)$, capturing alignment in shape and timing while being invariant to scale.

*3) Cross-coherence:* Cross coherence measures the frequency-specific linear relationship between original and reconstructed signals [20]. For each frequency $f$:

$$\mu_{x\hat{x}}(f) = \frac{|P_{x\hat{x}}(f)|^2}{P_{xx}(f)P_{\hat{x}\hat{x}}(f)}, \qquad (1)$$

where $P_{x\hat{x}}(f)$ is the cross-spectral density, and $P_{xx}(f)$, $P_{\hat{x}\hat{x}}(f)$ are the power spectral densities of $x(t)$ and $\hat{x}(t)$. To obtain a single summary metric, we compute the spectrum-weighted average coherence [21], [22]:

$$\bar{\mu}_{x\hat{x}} = \frac{\sum_f P_{xx}(f) \cdot \mu_{x\hat{x}}(f)}{\sum_f P_{xx}(f)}. \qquad (2)$$

This emphasizes coherence in frequency bands where the original signal carries more power.

### III. RESULTS

All analyses were performed using a segment window of $\Delta t = 0.5$ s. Alternative durations ($0.75$ s and $1$ s) were tested but did not improve model performance. The $0.5$ s window was thus selected for simplicity and efficiency, and all reported results correspond to this setting.

Table I summarizes the performance of the LASSO and MLP models under causal, anti-causal, and non-causal configurations for both ECG→PCG and PCG→ECG transformations in the within-subject setting.

Overall, the neural network consistently outperforms the linear LASSO model across all metrics and configurations. For example, in the non-causal ECG→PCG task, MLP achieves coherence $0.37 \pm 0.16$, correlation $\rho_{x\hat{x}} = 0.49 \pm 0.27$, and SNR $1.8 \pm 1.9$ dB. For PCG→ECG, performance improves further, with coherence $0.75 \pm 0.19$, $\rho_{x\hat{x}} = 0.78 \pm 0.20$, and SNR $6.5 \pm 5.2$ dB. This demonstrates the model's strength in capturing nonlinear temporal dependencies.

Transformation direction also impacts performance: PCG→ECG consistently yields higher metrics than ECG→PCG, indicating that reconstructing ECG from PCG is more effective—likely due to ECG's higher information content and lower noise.

Causality analysis reveals a consistent physiological pattern. For ECG→PCG, causal models outperform anti-causal ones (e.g., MLP coherence: 0.36 vs. 0.30), aligning with the expected direction of electrical-to-mechanical activity. In contrast, for PCG→ECG, anti-causal models perform better (e.g., MLP coherence: 0.71 vs. 0.59), suggesting that predicting electrical signals from subsequent mechanical responses is more effective in this setup.

Model comparisons highlight the value of nonlinearity. For ECG→PCG, LASSO performs comparably to MLP (e.g., non-causal coherence: 0.33 vs. 0.37), but for PCG→ECG, the performance gap is substantial (e.g., LASSO: 0.31 vs. MLP: 0.75), underlining the need for nonlinear models to effectively reconstruct ECG.

In summary, non-causal MLP models deliver the best overall performance for both transformation directions, providing the highest coherence, correlation, and SNR. The following sections further analyze the MLP's behavior and its implications for ECG-PCG cross-modal learning.

### A. Spectral Analysis

To assess frequency-domain fidelity, we compared the power spectra of original and reconstructed ECG and PCG

TABLE I: Evaluation metrics for LASSO and MLP models across causal, anti-causal, and non-causal approaches (cf. Fig. 2) in both ECG-to-PCG (ECG→PCG) and PCG-to-ECG (PCG→ECG) transformations.

| Translation Direction | State | Model | $\bar{\mu}_{\mathbf{x}\hat{\mathbf{x}}}$ | $\rho_{\mathbf{x}\hat{\mathbf{x}}}$ | SNR |
|---|---|---|---|---|---|
| ECG→PCG | Causal | MLP | 0.36 ± 0.15 | 0.47 ± 0.28 | 1.6 ± 1.9 |
| | | LASSO | 0.33 ± 0.13 | 0.38 ± 0.25 | 1.0 ± 1.4 |
| | Anti-causal | MLP | 0.30 ± 0.13 | 0.40 ± 0.23 | 1.0 ± 1.3 |
| | | LASSO | 0.29 ± 0.11 | 0.17 ± 0.10 | 0.2 ± 0.2 |
| | Non-causal | MLP | 0.37 ± 0.16 | 0.49 ± 0.27 | 1.8 ± 1.9 |
| | | LASSO | 0.33 ± 0.13 | 0.39 ± 0.26 | 1.1 ± 1.4 |
| PCG→ECG | Causal | MLP | 0.59 ± 0.19 | 0.69 ± 0.21 | 4.0 ± 3.3 |
| | | LASSO | 0.25 ± 0.13 | 0.20 ± 0.12 | 0.2 ± 0.3 |
| | Anti-causal | MLP | 0.71 ± 0.20 | 0.70 ± 0.25 | 5.2 ± 4.8 |
| | | LASSO | 0.29 ± 0.16 | 0.28 ± 0.18 | 0.5 ± 0.7 |
| | Non-causal | MLP | 0.75 ± 0.19 | 0.78 ± 0.20 | 6.5 ± 5.2 |
| | | LASSO | 0.31 ± 0.18 | 0.33 ± 0.20 | 0.7 ± 0.9 |

signals using the best-performing non-causal MLP and non-causal LASSO models (Fig. 3). This analysis highlights the strengths and limitations of nonlinear versus linear modeling.

For ECG reconstruction (Fig. 3a), the MLP output (red) closely matches the original spectrum (blue) across most frequencies, with notable underestimation below 4 Hz—reflecting ECG's distinctive low-frequency content seen in Fig. 1. In contrast, the LASSO output (yellow) shows substantial attenuation across all frequencies, particularly in the low range, confirming its limited spectral modeling capacity and aligning with its lower coherence and correlation in Table I.

For PCG reconstruction (Fig. 3b), the MLP captures better spectral content below 40 Hz and retains components up to 60 Hz, although with a faster high-frequency decay than the original. The LASSO model, by comparison, is largely limited to frequencies under 50 Hz and displays a 50 Hz notch—an artifact of preprocessing. This reflects the inherent constraints of linear models, which cannot generate spectral features absent in the input ECG, unlike nonlinear models like MLP that can reconstruct richer frequency content.

## IV. DISCUSSION

This study investigated information transfer and shared versus exclusive characteristics between ECG and PCG signals using the EPHNOGRAM dataset and a range of linear and nonlinear models. By analyzing causal, anti-causal, and non-causal architectures in the within-subject setting, we provide a detailed assessment of the potential and limitations of multimodal cardiac signal learning.

Nonlinear neural networks, particularly when applied in non-causal form, consistently outperformed linear approaches across all tasks and metrics. Their ability to capture complex temporal and spectral patterns makes them especially effective for waveform reconstruction, consistent with prior findings in multimodal cardiac modeling [4], [7], [23].

Spectral analyses further highlight the superiority of neural networks while revealing modality-specific challenges—such as reconstructing low-frequency ECG and high-frequency PCG components—where linear models like LASSO fall short, particularly for PCG→ECG mappings that require nonlinear transformations due to non-overlapping spectral content.

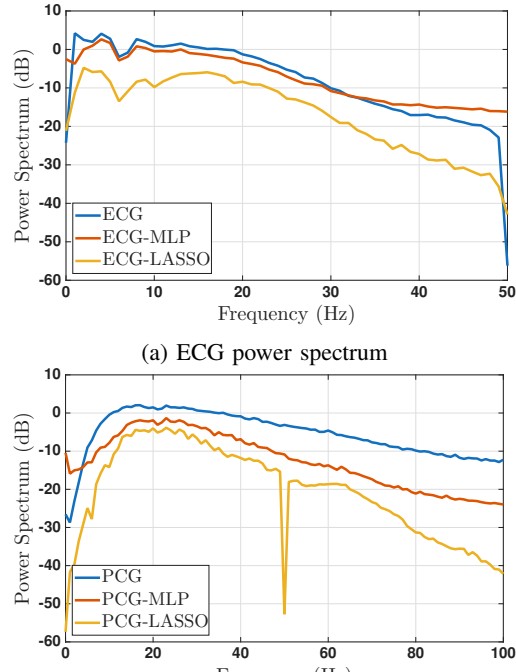

(a) ECG power spectrum

(b) PCG power spectrum

Fig. 3: Power spectra of original and reconstructed ECG and PCG signals using non-causal MLP and LASSO in the within-subject setting. Subplots (a) and (b) illustrate spectral alignment for each modality, showing that the nonlinear MLP more accurately captures the frequency content compared to the linear LASSO model.

A key finding is the asymmetry in transformation difficulty: reconstructing ECG from PCG is generally more accurate than the reverse, reflecting the physiological sequence where electrical activity (ECG) precedes mechanical response (PCG) [3]. Causality analysis supports this, showing that causal input windows benefit ECG→PCG, while anti-causal inputs better support PCG→ECG.

Exercise conditions significantly degrade reconstruction performance due to increased noise, motion artifacts, and rapidly changing cardiac dynamics. These results underscore the importance of robust preprocessing and adaptable models in real-

world, high-variability environments.

## A. Limitations and Future Work

We acknowledge some limitations in this work, which merit further research. The relatively small size of the EPHNO-GRAM dataset is a limitation;. At the same time, each subject provided 30-minute recordings during exercise and rest, which were substantial data to support our findings, larger datasets are required for population-level conclusions. Nonetheless, our current findings confirmed the causal relationship between electrical activity and mechanical contraction of the myocardium. For the same reason, we focused on more classical LASSO and MLP models and avoided more advanced and complex models like variational autoencoders (VAEs), which require more data. While we presented spectral-domain insights on the nonlinear interplay between the ECG and PCG, temporal importance analysis (attention weights or saliency over time) can further reveal how neighboring time-points and different phases of the cardiac cycle highly influence predictions. This study was within-subject; future research should explore cross-subject generalization, despite expected performance decline due to inter-individual differences and sensor variations. Incorporating biomarker validation, including QT interval and QRS duration estimation from reconstructed signals, is also essential for translating multimodal cardiac modeling into practical clinical and wearable monitoring applications. We will explore this direction of research in future work.

## V. Conclusion

This study presents a unified framework for modeling shared and exclusive information between ECG and PCG using data-driven multimodal learning. Non-causal neural network models consistently achieve the best performance, particularly for PCG to ECG transformations. Model accuracy is influenced by physiological state and inter-subject variability. Spectral analysis further confirms that nonlinear models are better equipped to preserve physiologically meaningful frequency content.

By demonstrating the feasibility of reconstructing clinically relevant ECG waveforms from PCG, this work advances the potential for multimodal cardiac monitoring in both clinical and resource-limited settings. Continued development of robust, generalizable, and interpretable machine learning models will be essential for translating these findings into practical applications for wearable and remote cardiac health monitoring.

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
