# OpenReview forum: "Bidirectional Translation Between ECG and PCG"
_IEEE.org/EMBS/BHI/2025/Conference — BHI 2025_

### Official Review · Reviewer_NoUJ · 2025-06-27
**Bidirectional Translation Between ECG and PCG**

**Confidence:** 4
**Clarity Of Writing:** great
**Clinical Significance:** good
**Methodological Novelty:** good
**Overall Rating:** 7

**Experiments And Results:**

great

**Questions For The Authors:**

Cross-Subject Generalization
Question for the authors: Have you explored or considered cross-subject modeling to evaluate how well your approach generalizes to unseen individuals? If not, could you clarify why this was excluded?
Impact: Demonstrating generalization is key for practical deployment. A clear explanation or additional experiments could significantly improve the perceived robustness and applicability of your method.

Clinical Relevance of Reconstructed Signals
Question: Have you evaluated whether clinically relevant features (e.g., QRS duration, QT interval, S1/S2 timing) are preserved in the reconstructed signals? If not, do you plan to include such analyses in future work?
Impact: Including even preliminary biomarker-level validation would greatly strengthen claims of clinical utility and could raise the significance score.

Choice of MLP Architecture
Question: Why was a relatively shallow MLP selected over more expressive architectures like CNNs or transformers, which are standard in biosignal modeling? Was this a deliberate trade-off for interpretability or resource constraints?
Impact: Justification for model choice or discussion of alternatives could clarify the paper’s scope and raise the novelty and methodological rigor score.

Interpretability of Nonlinear Models
Question: Did you attempt to interpret what features the neural network learned (e.g., via saliency maps or feature importance)? This could enhance the physiological understanding of ECG-PCG coupling.
Impact: Including interpretability insights would increase the scientific depth and may positively impact both novelty and impact scores.

**Strengths:**

Here are the key strengths:

Novel and comprehensive analysis of bidirectional ECG-PCG translation using both causal and non-causal frameworks.

Robust comparison of linear vs. nonlinear models, highlighting the superiority of neural networks for waveform reconstruction.

Thorough evaluation using multiple metrics (SNR, correlation, coherence) and spectral analysis, providing deep physiological insight.

Practical relevance for wearable and remote monitoring applications, supported by real-world dataset (EPHNOGRAM).

**Summary Of The Paper:**

The study explores the potential of reconstructing ECG from PCG and vice versa using linear (LASSO) and nonlinear (MLP) models, especially focusing on within-subject modeling under rest and exercise conditions using the EPHNOGRAM dataset. The paper emphasizes the superior performance of non-causal neural networks, particularly for PCG→ECG reconstruction.

**Weaknesses:**

Limited generalizability - The study focuses only on within-subject modeling; cross-subject performance is not evaluated.

Lack of clinical validation: No analysis of clinical biomarkers (e.g., QRS duration) from reconstructed signals.

Simplistic model architecture: The MLP used is relatively basic; more advanced architectures (e.g., CNNs, transformers) are not explored.

Missing interpretability tools: No effort to interpret learned features or model decisions, especially for the neural network.

---

### Official Review · Reviewer_gFT3 · 2025-07-16
**A well-motivated study exploring the bidirectional reconstruction of ECG and PCG waveforms using both linear and nonlinear models**

**Confidence:** 5
**Clarity Of Writing:** great
**Clinical Significance:** excellent
**Methodological Novelty:** excellent
**Overall Rating:** 8

**Experiments And Results:**

great

**Questions For The Authors:**

1. Have you considered using constrained representation learning methods such as variational autoencoders (VAEs), which are commonly applied in cross-domain or modality translation tasks? How might such approaches compare to your current framework?
2. You emphasize the importance of robust preprocessing for signal quality. Have you conducted any comparative analyses between models trained with and without preprocessing? Quantifying the impact of preprocessing on reconstruction accuracy would help clarify its contribution.

**Strengths:**

1. Novel framing of bidirectional translation between ECG and PCG signals
2. The inclusion of three temporal contexts (causal, anti-causal, and non-causal) alongside both linear and nonlinear models (LASSO and MLP) demonstrates a comprehensive exploration of the translation space with clear and systematic methodology.

**Summary Of The Paper:**

This paper investigates the feasibility of reconstructing electrocardiogram (ECG) and phonocardiogram (PCG) signals from each other. These two modalities reflect the electrical and mechanical activities of the heart, yet lack of research has examined the extent to which information is shared or exclusive between them. Leveraging the EPHNOGRAM dataset, the authors develop a unified cross-modal learning framework to assess bidirectional translation under three temporal settings: causal, anti-causal, and non-causal. They evaluate two model types—LASSO (linear) and MLP (nonlinear)—to examine both waveform reconstruction and spectral fidelity. Results indicate that nonlinear MLP models consistently outperform LASSO, with particularly strong performance in the PCG-to-ECG direction. The study concludes that nonlinear models can effectively capture intermodal cardiac relationships, even under noisy or real-world conditions such as during physical activity.

**Weaknesses:**

The dataset is relatively small and lacks population diversity (n=24), which may limit the generalizability of the findings to broader clinical or demographic groups.

---

### Official Review · Reviewer_o5Wp · 2025-07-18
**Bidirectional Translation Between ECG and PCG**

**Confidence:** 4
**Clarity Of Writing:** great
**Clinical Significance:** great
**Methodological Novelty:** great
**Overall Rating:** 7

**Experiments And Results:**

great

**Questions For The Authors:**

--

**Strengths:**

- The premise is very interesting and a fascinating problem.
- The explanation of the methods is comprehensive and provides the necessary details to justify the methodology
- The results are clearly explained and the explainations provided, especially for PCG -> ECG reconstruction are valid

**Summary Of The Paper:**

This paper explores the bidirectional translation between electrocardiography (ECG) and phonocardiography (PCG), aiming to understand the shared and unique information within these cardiac signals. Researchers utilize the EPHNOGRAM dataset, containing simultaneous ECG-PCG recordings from healthy adults during rest and exercise, to analyze the potential for reconstructing one signal from the other. Nonlinear neural network models, particularly a non-causal neural network, consistently demonstrate superior performance over linear models in this reconstruction, with PCG-based ECG reconstruction proving more feasible. The findings offer quantitative insights into the electromechanical relationship of cardiac signals, supporting the development of advanced multimodal cardiac monitoring tools.

**Weaknesses:**

- Can the small dataset provide conclusive relationships between the two modalities?